# PRISM: Diversifying Dataset Distillation by Decoupling Architectural Priors

**Brian Moser**[1], **Shalini Sarode**[1,2], **Federico Raue**[1], **Stanislav Frolov**[1],
**Krzysztof Adamkiewicz**[1,2], **Arundhati Shanbhag**[1,2], **Joachim Folz**[1],
**Tobias Nauen**[1,2], **Andreas Dengel**[1,2]

[1]**German Research Center for Artificial Intelligence (DFKI)**
[2]**RPTU Kaiserslautern-Landau**

`first.last@dfki.de`

## Abstract

Dataset distillation (DD) promises compact yet faithful synthetic data, but existing approaches often inherit the inductive bias of a single teacher model. As dataset size increases, this bias drives generation toward overly smooth, homogeneous samples, reducing intra-class diversity and limiting generalization. We present PRISM (PRIors from diverse Source Models), a framework that disentangles architectural priors during synthesis. PRISM decouples the logit-matching and regularization objectives, supervising them with different teacher architectures: a primary model for logits and a stochastic subset for batch-normalization (BN) alignment. On ImageNet-1K, PRISM consistently and reproducibly outperforms single-teacher methods (e.g., SRe2L) and recent multi-teacher variants (e.g., G-VBSM) at low- and mid-IPC regimes. The generated data also show significantly richer intra-class diversity, as reflected by a notable drop in cosine similarity between features. We further analyze teacher selection strategies (pre- vs. intra-distillation) and introduce a scalable cross-class batch formation scheme for fast parallel synthesis. Code: `https://github.com/Brian-Moser/prism`.

## 1 Introduction

Dataset distillation (DD) has emerged as a critical and controllable data generation method in modern deep learning, motivated by three core objectives: enhanced robustness against adversarial attacks (Lai et al., 2025), improved privacy through membership and model inversion safeguards (Dong et al., 2022; Carlini et al., 2022), and efficient data compression (Zhao et al., 2020; Zhao & Bilen, 2023). Unlike large generative visual models such as diffusion models, DD produces images whose class semantics are guaranteed by gradient supervision, *i.e.*, they produce training-signal-equivalent samples (Dhariwal & Nichol, 2021; Fort & Whitaker, 2025; Cazenavette et al., 2022).

As DD techniques have matured, a significant challenge remains unresolved: the failure to synthesize diverse, large-scale datasets that capture the complexity of their real-world counterparts. Current DD methods, including gradient matching (Yin et al., 2023) or parameter matching (Cazenavette et al., 2022; Cui et al., 2023), also in combination with generative priors (Cazenavette et al., 2023; Moser et al., 2024; Su et al., 2024; Duan et al., 2023), predominantly concentrate on high-resolution synthetic images in small-scale settings, such as 50 or 100 images-per-class (IPC) and fail to close the gap between the full and compressed datasets.

In this work, we question the motivation of compressing a dataset, since *memory is cheap*, in favor of the other motivations, namely robustness and privacy. Yet, naively upscaling DD techniques for an identical dataset size often results in overly smooth, feature-restricted datasets lacking sufficient diversity (Shao et al., 2024a; Sun et al., 2024; Shen et al., 2025), as illustrated in Figure 1. This leads to *homogeneous representations devoid of sufficient intra-class variability*, thus impairing robustness as well as safety and ultimately limiting the practical utility of DD (Sorscher et al., 2022).

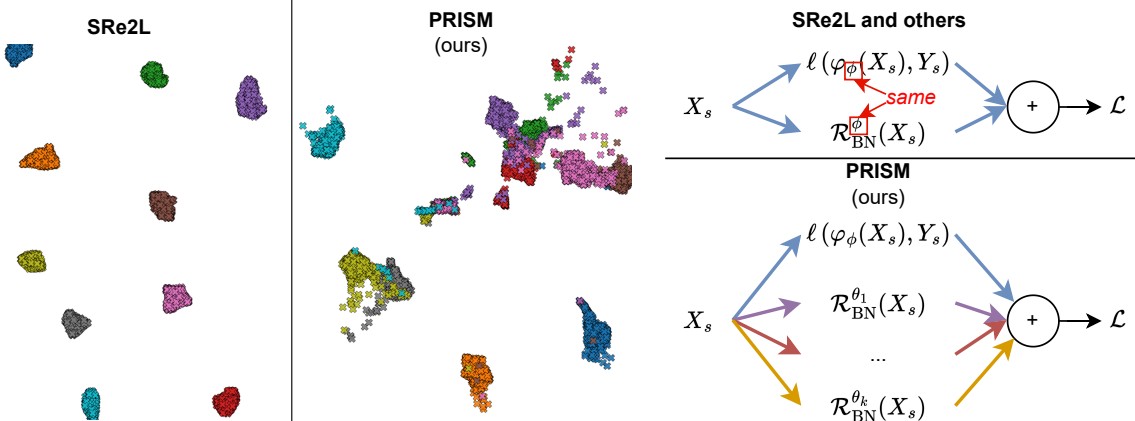

Figure 1: UMAP visualization of synthetic images from ImageNet-1K (10 classes), comparing SRe2L with our proposed multi-teacher alignment. Our approach, PRISM, generates significantly greater intra-class diversity, contrasting the overly uniform clusters of SRe2L that can lead to model overfitting more easily.

Figure 2: The core idea behind PRISM (**PRI**ors from diverse **S**ource **M**odels): Use multiple, diverse models for decoupling the logit maximization and regularization through BN alignment instead of one, like in SRe2L and related work.

Our central argument is that knowledge within a trained network is inseparable from the architecture that contains it (Ulyanov et al., 2018; Shao et al., 2024a). Any single model possesses a strong inductive bias, *i.e.*, its "view" of the world. Distilling a dataset through one model inevitably imprints this single, limited view onto the synthetic data, resulting in a homogenous dataset that fails to train generalizing models. *In order to create a truly generalizable synthetic dataset, we must synthesize it from a distribution of "world views".*

Addressing this limitation, we propose diversifying the distillation process through a multi-architectural prior framework that simultaneously optimizes logit matching (Yin et al., 2023) and regularization through batch normalization (BN) alignment (Yin et al., 2020) with at least two decoupled teacher models, which we coin PRISM (**PRI**ors from diverse **S**ource **M**odels). As such, our teacher decoupling is orthogonal to all existing scaling attempts, as shown in Figure 2. Instead of relying on training schedules (Shen et al., 2025), data initialization (Sun et al., 2024), or post-evaluation pipelines (Shao et al., 2024b), PRISM introduces an orthogonal mechanism by *decoupling the logit-matching objective from the BN-alignment regularization*. This allows different architectural priors - from multiple, distinct teacher models - to simultaneously contribute to different aspects of the synthesis, complementing rather than replacing prior innovations.

In summary, our contributions are clear and well-scoped within the ongoing evolution of dataset distillation:

- First, we introduce PRISM, a novel framework that tackles the lack of diversity in dataset distillation by decoupling the architectural priors used for logit supervision and BN-alignment regularization.

- Second, we provide a systematic analysis of teacher-selection strategies, demonstrating that a pre-distillation selection of diverse teachers is highly effective.

- Third, we show that PRISM not only sets new state-of-the-art results, achieving up to **70.4%** top-1 accuracy with a ResNet-101 at IPC=100, but also generates datasets with quantifiably greater intra-class diversity, directly addressing the critical challenge of homogeneity in modern dataset distillation, while still maintaining a simple and massively parallelizable synthesis pipeline that scales efficiently to large datasets like ImageNet-1K.

## 2 Related Work

**Dataset Distillation.** The field has largely bifurcated into two main strategies. Early and prominent methods relied on *batch-to-batch matching*, which involves expensive bi-level optimization to align gradients (Zhao et al., 2020), training trajectories (Cazenavette et al., 2022; Cui et al., 2023), or feature distributions (Zhao & Bilen, 2023) between real and synthetic data batches. While effective, the computational overhead of these methods makes them challenging to apply to large-scale datasets.

Thus, the community has shifted towards *batch-to-global matching*, a paradigm pioneered by SRe2L (Yin et al., 2023). This approach uses a pre-trained teacher model to generate global statistics (*e.g.*, from batch normalization layers) and then optimizes synthetic images to match these global targets. This strategy is significantly more efficient and has enabled distillation at the scale of ImageNet-1K. However, because every synthetic image in a class is optimized against the same global signal, this approach often suffers from a lack of intra-class diversity (Shao et al., 2024a).

**Strategies for Enhancing Diversity.** Recognizing the diversity challenge in *batch-to-global matching*, several methods have proposed alternative solutions. **G-VBSM** (Shao et al., 2024a) was a key step in this direction, introducing the idea of using multiple teacher models and statistical metrics (beyond just BN statistics) to create a richer, more varied supervision signal. **EDC** (Shao et al., 2024b) further refined this multi-teacher approach by identifying a suite of critical, and often overlooked, design choices in the post-training and evaluation pipeline, such as smoothing learning rate schedules and EMA-based evaluation, that are necessary to unlock the full potential of a diverse distilled dataset.

Another related direction is **D3S** (**?**), which employs an ensemble of independently trained ResNet-18 teachers and rotates them during optimization to reduce single-model bias. While similar in spirit, it keeps a fixed architecture family and couples the same ensemble for both soft-labeling and feature-matching, whereas PRISM decouples *both the teacher roles and the architectural priors*, enabling a richer and more diverse supervision signal. **CV-DD** (**?**) improves diversity by letting a committee of heterogeneous teachers vote on soft labels, reducing the bias of any single model. Unlike PRISM, which diversifies the *training signal* by decoupling logit and BN supervision during synthesis, CV-DD focuses on the *target labels* rather than the optimization dynamics themselves. As such, it is complementary but orthogonal to our architectural decoupling strategy.

Other methods have approached diversity from different angles. **RDED** (Sun et al., 2024) focuses on the initialization data itself, using multi-crop image concatenation to create varied starting points for distillation. While this improves performance and speed, it sidesteps the goal of generating purely synthetic data, as the resulting images are composites of real images, which may not satisfy the privacy and robustness motivations of DD. In contrast, **DELT** (Shen et al., 2025) introduces the "EarlyLate" training scheme, where images are synthesized for varying numbers of iterations to create a spectrum of synthetic samples - from realistic to abstract.

## 3 Methodology

In the following, we will derive the method behind PRISM step-by-step, starting from classical DD and SRe2L. Consider a real dataset $\mathcal{T} = (X_r, Y_r)$ comprising $N$ images, where $X_r \in \mathbb{R}^{N \times H \times W \times C}$ are the real images. DD aims to distill this dataset into a smaller synthetic set $\mathcal{S} = (X_s, Y_s)$, where $X_s \in \mathbb{R}^{M \times H \times W \times C}$ with $M \ll N$. Conventionally, $M = \mathcal{C} \cdot \text{IPC}$, where $\mathcal{C}$ is the number of classes and IPC are the specified images-per-class. Formally, classical DD seeks the optimal synthetic dataset $\mathcal{S}^*$ that minimizes the distillation loss $\mathcal{L}(\mathcal{S}, \mathcal{T})$:

$$\mathcal{S}^* = \arg\min_{\mathcal{S}} \mathcal{L}(\mathcal{S}, \mathcal{T}). \tag{1}$$

Here, we follow the formulation of SRe2L (Yin et al., 2023) by optimizing over an output of a teacher model $\varphi_\phi$ with parameters $\phi$:

$$X_S^* = \arg\min_{X_S} \ell\left(\varphi_\phi(X_s), \boldsymbol{Y}_s\right) + \lambda \mathcal{R}_{\text{reg}}, \tag{2}$$

where $\mathcal{R}_{\text{reg}}$ is the regularization term to avoid noise-like artifacts, *i.e.*, regularize synthetic images to look more natural, and $\lambda$ is its weighting hyperparameter. While there are multiple valuable options for $\mathcal{R}_{\text{reg}}$, such as L2 or TV regularization, the authors of SRe2L found that using the deep inversion (Yin et al., 2020) inspired BN alignment $\mathcal{R}^{\boldsymbol{\theta}}_{\text{BN}}$ of the model alone led to the best overall performance:

$$\mathcal{R}_{\text{reg}} = \mathcal{R}^{\boldsymbol{\theta}}_{\text{BN}}(X_s) = \sum_l \left\| \mu_{l,\boldsymbol{\theta}}(X_s) - \mathbb{E}\left(\mu_{l,\boldsymbol{\theta}} \mid \mathcal{T}\right) \right\|_2 + \sum_l \left\| \sigma^2_{l,\boldsymbol{\theta}}(X_s) - \mathbb{E}\left(\sigma^2_{l,\boldsymbol{\theta}} \mid \mathcal{T}\right) \right\|_2, \tag{3}$$

where $l$ is the index of BN layer in the model with parameters $\theta$, $\mu_{l,\boldsymbol{\theta}}(\widetilde{\boldsymbol{x}})$ and $\sigma^2_{l,\boldsymbol{\theta}}(\widetilde{\boldsymbol{x}})$ are mean and variance, which can be conveniently approximated by the running mean and running variance in a pre-trained model at the $l$-th layer.

### 3.1 Dual-Teacher Decoupling

The standard SRe2L framework (Yin et al., 2023) uses a single, pre-trained teacher model for both parts of the objective function. This means the same model architecture and weights provide the supervision for both the logit-matching term (governed by parameters $\boldsymbol{\phi}$) and the BN-alignment regularization (which depends on the BN layer parameters within $\boldsymbol{\theta}$). We refer to this standard approach, where a single model fulfills both roles, as *single-teacher alignment*. Thus, $\boldsymbol{\phi} = \boldsymbol{\theta}$.

The core idea of PRISM is to challenge this coupling. We propose to *decouple* the architectural priors by allowing different models to supervise each term, a strategy we coin *multi-teacher alignment*. In this setup, the logit teacher's parameters ($\boldsymbol{\phi}$) and the BN teacher's parameters ($\boldsymbol{\theta}$) belong to distinct models. For instance, one could use an EfficientNet as the logit teacher and a standard ResNet as the BN teacher. This leads to an optimization where the gradient is a composite of two different architectural perspectives:

$$\nabla_{X_s}\mathcal{L}(\mathcal{S}, \mathcal{T}) = \underbrace{\nabla_{X_s}\ell(\varphi_\phi(X_s), Y_s)}_{\text{Teacher 1}} + \lambda \underbrace{\nabla_{X_s}\mathcal{R}^\theta_{BN}(X_s)}_{\text{Teacher 2}}$$

As a result, the optimization of $X_s$ is guided by two distinct and potentially complementary objectives derived from different architectural priors:

- $\nabla_{X_s}\ell(\varphi_\phi(...))$: This term pushes $X_s$ to have features that are effective for classification from the perspective of the logit teacher. If optimized in isolation, this objective is known to produce adversarial-like patterns that lack semantic realism and, therefore, lead to poor generalization (Yin et al., 2020).

- $\nabla_{X_s}\mathcal{R}^\theta_{BN}(...)$: This term pushes $X_s$ to have low-level global feature statistics (mean and variance) that are considered "natural" from the perspective of the BN teacher model with parameters $\theta$, a countermeasure against adversarial-like patterns.

Unlike prior ensemble-based methods (*e.g.*, G-VBSM (Shao et al., 2024a), D3S (**?**), and CC-DD (**?**)), which aggregate identical architectures within a single objective, PRISM introduces a structural decoupling: different architectures supervise separate loss terms (*i.e.*, logit matching and BN regularization). This decoupling changes the optimization dynamics, enabling complementary architectural priors rather than redundant ones.

### 3.2 Generalized Multi-Teacher Alignment

As a natural next step, a more generalized version of our proposed decoupling is to apply multiple models for BN alignment. Thus, we obtain **PRI**ors from diverse **S**ource **M**odels (**PRISM**). More concretely, let $\mathcal{M} = \{\varphi_{\boldsymbol{\theta}_1}, \ldots, \varphi_{\boldsymbol{\theta}_k}\}$ be $k$ models used for BN alignment, the objective becomes:

$$X_S^* = \arg\min_{X_S} \ell\left(\varphi_{\boldsymbol{\phi}}(X_s), \boldsymbol{Y}_s\right) + \lambda\mathcal{R}_{\text{reg}}$$
$$= \arg\min_{X_S} \ell\left(\varphi_{\boldsymbol{\phi}}(X_s), \boldsymbol{Y}_s\right) + \lambda \sum_{\boldsymbol{\omega} \in \mathcal{M}} \mathcal{R}^{\boldsymbol{\omega}}_{\text{BN}}(X_s) \tag{4}$$

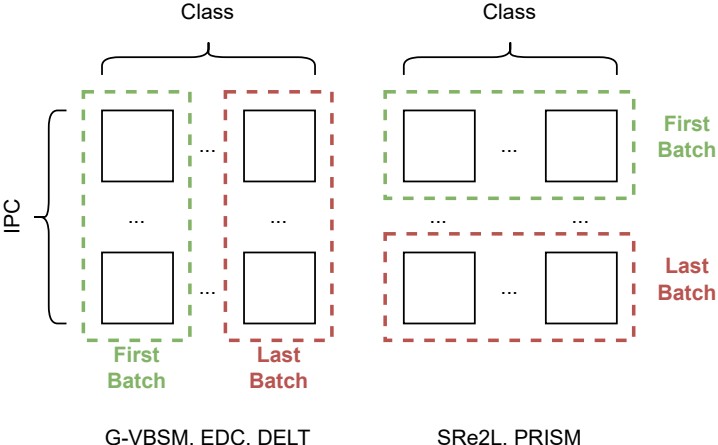

Figure 3: Batch formation and optimization strategies. (Left) Methods like G-VBSM, EDC, and DELT optimize jointly over all classes simultaneously. (Right) Methods like our PRISM and SRe2L process each IPC index independently.

This core principle is visualized in Figure 2. To further boost diversity, we recommend including probabilities of using more than one BN alignment for each distilled image. To formalize this, let $\mathcal{M}_{\text{pool}} = \{\varphi_{\boldsymbol{\theta}_1}, \ldots, \varphi_{\boldsymbol{\theta}_{k_{\text{total}}}}\}$ be the full set of $k_{\text{total}}$ available source models for BN alignment, and let $k_{\max}$ be the maximum number of BN teachers to use simultaneously, constrained by VRAM. We define the set of all valid, non-empty subsets of teachers as $\mathbb{M}_{\text{valid}} = \{\mathcal{M}_{\text{sub}} \subseteq \mathcal{M}_{\text{pool}} \mid 1 \leq |\mathcal{M}_{\text{sub}}| \leq k_{\max}\}$.

Thus, we sample a subset $\mathcal{M}_{\text{sub}}$ from a distribution $P$ over all possible valid subsets, $\mathcal{M}_{\text{sub}} \sim P(\mathbb{M}_{\text{valid}})$. A simple and effective choice for $P$ is the uniform distribution. The overall objective is then to minimize the expected loss over this random selection:

$$X_S^* = \arg\min_{X_S} \ell\left(\varphi_{\boldsymbol{\phi}}(X_s), \boldsymbol{Y}_s\right) + \mathbb{E}_{\mathcal{M}_{\text{sub}} \sim P(\mathbb{M}_{\text{valid}})} \left[\lambda \sum_{\boldsymbol{\omega} \in \mathcal{M}_{\text{sub}}} \mathcal{R}_{\text{BN}}^{\boldsymbol{\omega}}(X_s)\right] \tag{5}$$

By following the insights of Tran et al. (2021), we further claim improved robustness by this generalized multi-teacher alignment by providing a proof sketch in the appendix.

### 3.3 Teacher-Selection Strategy

To further dissect the role of teacher model selection in distillation, we explore two distinct teacher-selection strategies: (1) a *pre-distillation selection* strategy, in which a fixed set of teacher models is determined before the distillation begins, and (2) an *intra-distillation selection* strategy, where teachers are dynamically selected during the distillation process itself.

More specifically, in the *pre-distillation* strategy, the set of active teachers is determined once for each synthetic image before the optimization process begins. For a given image $X_s$, we sample a single gradient teacher $\varphi_{\boldsymbol{\phi}}$ and a corresponding subset of BN alignment teachers $\mathcal{M}_{\text{sub}}$. This fixed ensemble then guides the entire distillation process for that specific image.

Conversely, the *intra-distillation* strategy leads to a more dynamic distillation process by re-selecting teachers *during* the optimization itself, a method heavily inspired by G-VBSM (Shao et al., 2024a). Within PRISM, this means that at each distillation step, a new set of teachers, both the gradient teacher $\varphi_{\boldsymbol{\phi}}$ and the BN alignment subset $\mathcal{M}_{\text{sub}}$, can be re-sampled from their respective pools.

Table 1: Comparison with state-of-the-art dataset distillation on **ImageNet-1K**. We report top-1 accuracy [%] for ResNet-18/50/101 trained on distilled datasets at IPC = 10, 50, 100 reporting mean±std over three seeds. While competitive at low IPCs, our method, PRISM, consistently and reproducibly establishes new state-of-the-art performance at higher IPCs (50 and 100) across all architectures and evaluation protocols. Results marked with ‡ use DELT's evaluation procedure (Shen et al., 2025). Bold indicates the best value per column. All other results follow our validation protocol. "−" denotes not reported.

| Method | ResNet-18 | | | ResNet-50 | | | ResNet-101 | | |
|---|---|---|---|---|---|---|---|---|---|
| | (IPC=10) | (IPC=50) | (IPC=100) | (IPC=10) | (IPC=50) | (IPC=100) | (IPC=10) | (IPC=50) | (IPC=100) |
| SRe2L | 21.3±0.6 | 46.8±0.2 | 52.8±0.3 | 28.4±0.1 | 55.6±0.3 | 61.0±0.4 | 30.9±0.1 | 60.8±0.5 | 62.8±0.2 |
| G-VBSM | 31.4±0.5 | 51.8±0.4 | 55.7±0.4 | 35.4±0.8 | 58.7±0.3 | 62.2±0.3 | 38.2±0.4 | 61.0±0.4 | 63.7±0.2 |
| RDED | 42.0±0.1 | 56.5±0.1 | 59.8±0.1 | - | - | - | 30.9±0.1 | 60.8±0.5 | 62.8±0.2 |
| EDC | 48.6±0.3 | 58.0±0.2 | - | **54.1±0.2** | 64.3±0.2 | - | **51.7±0.3** | 64.9±0.2 | - |
| **PRISM** | **49.4±0.2** | **59.0±0.1** | **60.9±0.2** | 51.1±1.2 | **65.1±0.1** | **67.5±0.2** | 48.5±1.7 | **65.9±0.2** | **68.6±0.4** |
| DELT ‡ | 46.1±0.4 | 59.2±0.4 | 62.4±0.2 | - | - | - | **48.5±1.6** | 66.1±0.5 | 67.6±0.3 |
| **PRISM ‡** | **46.9±0.1** | **59.6±0.2** | **62.7±0.1** | **46.3±0.7** | **66.5±0.2** | **69.4±0.1** | 40.7±3.6 | **66.7±0.2** | **70.4±0.2** |

## 3.4 Batch Formation and Parallelization Strategy

A key design choice that distinguishes PRISM from other recent methods like G-VBSM (Shao et al., 2024a), EDC (Shao et al., 2024b), and DELT (Shen et al., 2025) is the batch formation strategy during data synthesis. PRISM, following SRe2L, processes each image-per-class (IPC) index independently. This creates *cross-class* batches where each batch consists of a single IPC slice across multiple classes (*e.g.*, the $i$-th image from each class). The primary advantage of this strategy is its high efficiency and straightforward parallelizability; the synthesis of each IPC can be treated independently and easily distributed across multiple GPUs.

In contrast, other approaches often form *intra-class* batches, which contain multiple images from the same class, as shown in Figure 3. This enables specific regularization, such as the data densification in G-VBSM/EDC or diversity-driven optimization in DELT, which operates on the same-class images within a batch. While this improves diversity, this also comes at the cost of increased complexity, as such regularizations introduce intra-batch dependencies during optimization, *e.g.*, through explicitly pushing images during distillation apart. Our method prioritizes a simple and massively parallelizable pipeline, achieving diversity not through complex intra-batch information exchange between images, but through our core contribution of BN decoupling and diversifying architectural priors.

## 4 Experiments

### 4.1 Setup

We follow the well-established SRe2L pipeline (Yin et al., 2023) and its standard configurations for the recovery stage to ensure methodological continuity. Moreover, we initialize the synthetic dataset by selecting each real image exactly once from the ImageNet training set, ensuring a direct and consistent comparison across different configurations in our identical dataset-size distillation (no coreset selection like in DELT or multi-image initialization as in RDED/EDC). During the distillation process, we evaluate multiple teacher models to diversify synthetic data representations. We set a maximum of 4000 optimization iterations.

While most dataset distillation papers include CIFAR-10 results, we deliberately focus on ImageNet-1K and CIFAR-100 for two reasons. First, PRISM is designed to address the architectural bias and diversity limitations that become pronounced only in large-scale, high-class-count settings. Since the goal of PRISM is to *scale diversity*, ImageNet-1K and CIFAR-100 is the minimal scale at which its contributions are meaningful. We emphasize that all baselines are reproduced and compared under identical conditions, ensuring fair benchmarking at the appropriate scale.

Table 2: Comparison with state-of-the-art dataset distillation methods on **CIFAR-100** using **ResNet-18**. We report top-1 accuracy [%] at IPC = 1, 10, 50. PRISM (ours) will be filled in once evaluated.

| IPC | SRe2L | G-VBSM | RDED | EDC | PRISM (ours) |
|---|---|---|---|---|---|
| 1 | $2.0 \pm 0.2$ | $25.9 \pm 0.5$ | $11.0 \pm 0.3$ | $39.7 \pm 0.1$ | $\mathbf{41.5 \pm 0.2}$ |
| 10 | $31.6 \pm 0.5$ | $59.5 \pm 0.4$ | $42.6 \pm 0.2$ | $63.7 \pm 0.3$ | $\mathbf{64.7 \pm 0.2}$ |
| 50 | $49.5 \pm 0.3$ | $65.0 \pm 0.5$ | $62.6 \pm 0.1$ | $68.6 \pm 0.2$ | $\mathbf{69.4 \pm 0.3}$ |

## 4.2 Classical Dataset Distillation

Table 1 summarizes our main results, offering a balanced comparison between PRISM and prior state-of-the-art methods on ImageNet-1K. Table 2 summarizes our results on CIFAR-100. To ensure a fair and comprehensive analysis on ImageNet-1K, we present results under two distinct evaluation protocols: our own, which is optimized for low-to-mid IPCs (more details follow in Section 4.5), and the protocol used by DELT, which excels at higher IPCs.

**Performance with PRISM's Optimized Evaluation.** Under our primary evaluation protocol, PRISM establishes new SOTA results compared to EDC in Table 1. While EDC shows strong performance at IPC=10 on larger backbones, PRISM consistently and reproducibly outperforms all prior methods at IPC=50 and IPC=100 across all tested architectures. For instance, on ResNet-18, PRISM achieves an accuracy of **49.4% at IPC=10**, surpassing EDC (48.6%), and extends its lead at **IPC=50 (59.0%)** and **IPC=100 (60.9%)**. This trend holds for larger models, where PRISM's performance of **65.1% (ResNet-50, IPC=50)** and **68.6% (ResNet-101, IPC=100)** confirms that decoupling architectural priors is a highly effective strategy for generating diverse and generalizable synthetic data.

**Performance with DELT's Evaluation Procedure.** When evaluated using the DELT protocol, PRISM's advantages become even more pronounced for mid-to-high IPC scenarios, setting new SOTA results across multiple settings in Table 1. For ResNet-18, PRISM outperforms DELT at **IPC=50 (59.6%)** and **IPC=100 (62.7%)**. The benefits of our diverse architectural priors scale to larger models, where PRISM achieves a remarkable **69.4% accuracy on ResNet-50** and **70.4% on ResNet-101** at IPC=100. PRISM's ability to excel under an evaluation pipeline tailored for a different method underscores the fundamental quality of the dataset it generates, proving its robustness across varied training configurations, especially for larger IPCs.

**Performance on CIFAR-100.** On CIFAR-100, we observe a similar trend when evaluating a ResNet-18 student, as summarized in Table 2. PRISM consistently outperforms prior methods across all IPC regimes, improving over EDC by up to **1.8** points at IPC=1 and maintaining a solid margin at IPC=10 and IPC=50.

## 4.3 Analysis on Diversity

Current dataset distillation methods, particularly those building upon SRe2L (Yin et al., 2023), heavily rely on post-recovery strategies such as Knowledge Distillation (Qin et al., 2024; Li et al., 2025) and highly specialized validation settings to achieve their reported performance (Shao et al., 2024b; Shen et al., 2025; Yin & Shen, 2023). To ensure an unbiased diversity assessment of our proposed architectural decoupling, and to isolate its direct impact on synthesized data diversity, we intentionally diverge from these post-processing techniques in this analysis (contrasting the setting in the previous setting).

Our rationale is that if increased diversity, stemming from our multi-architectural priors, is a truly orthogonal axis for scaling DD, then its pronounced performance improvement should be significantly observable even in this large-scale, simplified setting, free from the confounding factors of post-recovery optimization. Experimental details are outlined in the Appendix C.

Table 3: Study of **teacher alignment and selection** strategies on **ImageNet-1K**, **IPC≈1200**: final validation accuracy [%] of **ResNet-18**. In addition, we report the max. VRAM consumption during distillation under a distillation batch size of 100. Note that this is the result for recovery-only (no knowledge distillation).

| Variant | Model Selection | BN Teachers | Acc. [%] | VRAM [GB] |
|---|---|---|---|---|
| Baseline (real data) | - | - | 70.0 | - |
| SRe2L | - | 1 | 17.9 | 6.5 |
| + single-teacher alignment | intra-distillation | 1 | 18.3 | 6.5 |
| + dual-teacher decoupling | intra-distillation | 1 | 19.0 | 13.0 |
| + single-teacher alignment | pre-distillation | 1 | 32.4 | 6.5 |
| **+ dual-teacher decoupling** | **pre-distillation** | **1** | **36.2** | **13.0** |
| + multi-teacher alignment | pre-distillation | 2 | 37.4 | 18.5 |
| + multi-teacher alignment | pre-distillation | 3 | 38.7 | 26.0 |
| **+ multi-teacher alignment** | **pre-distillation** | **4** | **39.1** | **32.5** |

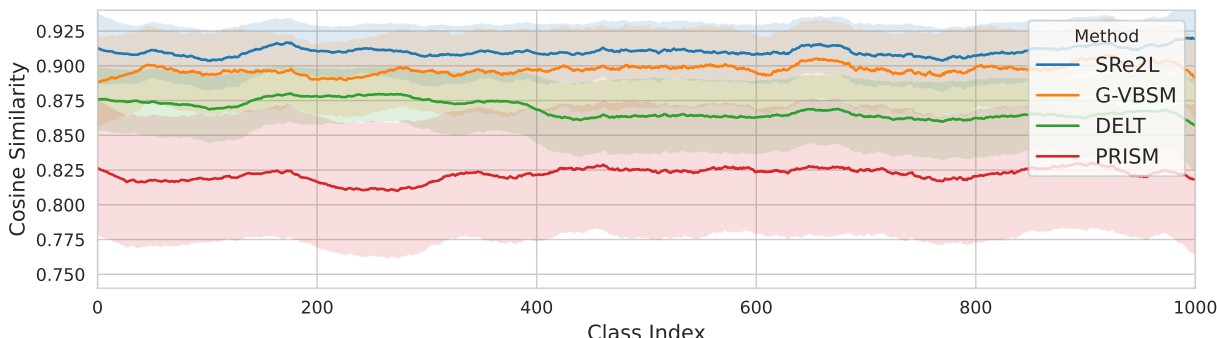

Figure 4: **Intra-class semantic cosine similarity** with a pretrained ResNet-18 model on ImageNet-1K dataset applied on the respective distilled images showing higher diversity as indicated by lower mean values and higher variance.

## 4.4 Analysis on Recovery-Only Diversity

Recent advances in dataset distillation increasingly couple the core data synthesis process with powerful post-recovery optimizations, such as knowledge distillation (Qin et al., 2024; Li et al., 2025) and highly specialized validation schedules (Shao et al., 2024b; Shen et al., 2025; Yin & Shen, 2023). While effective at boosting final accuracy, this entanglement can obscure the intrinsic quality of the generated data, making it difficult to attribute performance gains directly to the synthesis method itself. Therefore, to perform a rigorous and unbiased assessment of our architectural decoupling, this analysis intentionally isolates the synthesis stage from these downstream optimizations. Our rationale is that a truly effective diversity-enhancing method like PRISM demonstrates a clear advantage in this controlled setting, directly linking its architectural design to the quality of the distilled data. Experimental details for this setup without knowledge distillation are outlined in the appendix.

Table 3 summarizes the results of our recovery-only setting, from which we derive the following three positive observations: **(i)** Sampling multiple teacher models during distillation helps to improve the performance, but pre-selecting them for the whole distillation process is better. **(ii)** Decoupling gradient matching and BN statistics alignment further improves the performance. **(iii)** The best performance is achieved by using multiple models for BN statistics alignment. Taken together, all best performing configurations form our proposed method PRISM, namely multi-teacher alignment with 4BN priors and pre-distillation selection.

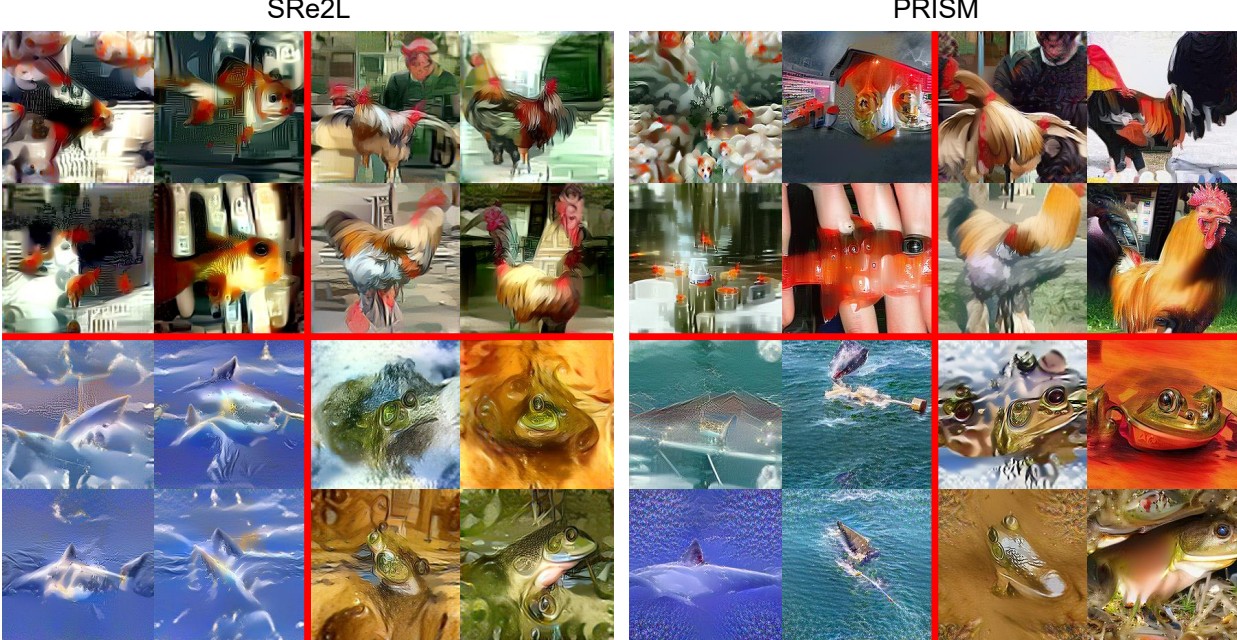

Figure 5: **Qualitative comparison of synthetic images** from ImageNet-1K generated by SRe2L and PRISM. Both methods start from the *exact same* initial real images to ensure a fair comparison. The images generated by SRe2L **(left)** exhibit significant homogeneity, with samples within each class (goldfish, rooster, shark, frog) converging to similar colors and textures. In contrast, PRISM **(right)** produces a wider variety of contexts and colorations.

To quantitatively validate that our method generates a more diverse dataset, we compute the intra-class semantic cosine similarity. This metric measures the average feature similarity between all pairs of synthesized images within the same class, using a pretrained ResNet-18 as a feature extractor; a lower similarity score thus indicates higher intra-class diversity. Figure 4 shows a clear separation between the methods. While existing approaches like SRe2L, G-VBSM, and DELT produce highly similar images (between 0.86 and 0.92), PRISM consistently achieves the lowest cosine similarity across all classes by a significant margin (mean values of 0.83 and below). This result provides strong quantitative evidence for our central claim: by decoupling and diversifying architectural priors, PRISM effectively breaks the homogeneity constraint inherent in single-teacher distillation, leading to the superior downstream performance reported in Table 1.

This quantitative evidence is further supported by a direct qualitative comparison, as shown in Figure 5. To ensure a fair assessment, both PRISM and SRe2L start from the exact same initial images. While SRe2L consistently produces homogeneous images where samples from the same class converge on similar textures and poses, PRISM generates a visibly more diverse set.

During this study, we also experiment with alternative regularization approaches besides BN alignment, which can be found in the appendix. In more detail, we try multi-resolution synthesis as an alternative regularization to BN alignment with CLIP embeddings (see Appendix D) as well as using large-scale pre-trained generative text-to-image models for identical dataset-size generation (see Appendix E).

### 4.5  Recovery and Post-Recovery Strategies

To identify the optimal configuration for PRISM, we conduct a systematic analysis beyond DELT's evaluation procedure, beginning from the vanilla SRe2L baseline. Our analysis starts with the generation of soft labels, then progresses to the image recovery process, and concludes with teacher model alignment and learning rate

Table 4: Study of **relabeling** and additional **recovery** strategies on **ImageNet-1K** with **ResNet-18**, **IPC=10**: final validation accuracy [%].

(a) **Soft-Label Targets**

| Variant | Acc. [%] |
|---|---|
| ResNet-18 Only | 21.22 |
| Ensemble Average | 23.75 |
| + MAE (GT=0.05) | 28.97 |
| + MSE (GT=0.05) | 28.07 |
| + MAE (GT=0.1) | 23.85 |
| **+ MSE (GT=0.1)** | **29.53** |

(b) **Add. Recovery Strategies**

| Variant | Acc. [%] |
|---|---|
| Baseline from (a) | 29.53 |
| + Resize Schedule | 29.93 |
| **+ Var. Iterations** | **31.30** |
| + Both | 30.89 |

(c) **Batch Size (Relabeling)**

| Batch Size | Acc. [%] |
|---|---|
| 1024 from (b) | 31.30 |
| 16 | 40.41 |
| 32 | 43.40 |
| **50** | **44.04** |
| 64 | 43.79 |
| 128 | 42.59 |

(d) **Teacher Strategies**

| Variant | Acc. [%] |
|---|---|
| Baseline from (c) | 44.04 |
| $\phi = \theta_1 \neq \theta_{>1}$ | **45.73** |

(e) **LR (Recovery)**

| LR | Acc. [%] |
|---|---|
| 0.25 from (d) | 45.73 |
| 0.10 | 46.63 |
| **0.05** | **47.35** |

(f) **Backbones**

| Model Pool | Acc. [%] |
|---|---|
| Models from (e) | 47.35 |
| - EfficientNet (Rel.) + AlexNet (Rel.) | 47.53 |
| **+ AlexNet (Rec.)** | **47.77** |

schedule refinements. We employ ResNet18, ResNet34, ShuffleNetV2-0.5, MobileNetV2, and EfficientNet-B0 during multi-teacher selection, using ResNet18 for logit maximization to ensure comparability with G-VBSM.

**Optimizing Soft-Label Generation.** Our first step is to refine the soft-label targets used for distillation. We find that moving beyond a single relabeler to an ensemble of the recovery models significantly improves performance, an idea inspired by G-VBSM (Shao et al., 2024a). As shown in Table 4a, this ensemble approach is most effective when using a Mean Squared Error (MSE) loss combined with a 0.1 ground-truth (GT) addition, which proves superior to both MAE and standard KL divergence.

**Refining the Recovery Process.** With improved soft labels, we turn our attention to the recovery stage itself. We investigate two key strategies: scheduling the augmentation strength and varying the number of distillation iterations. Although adjusting the minimum crop size of the random resize augmentation provides only a minor improvement, varying the number of distillation iterations per image, following DELT (Shen et al., 2025), yields the best performance (Table 4b).

**Batch Size during Relabeling.** With an effective loss function established, we next examine the batch size used during this relabeling phase. Our experiments, summarized in Table 4c, reveal that a batch size of 50 is optimal, aligning with similar findings in recent literature (Yin & Shen, 2023; Shao et al., 2024b). Even with distilled data, the relabeling process remains highly sensitive to batch statistics.

**First BN teacher.** Finally, we address the composition of the teacher models themselves. Confirming a key insight from SRe2L (Yin et al., 2023), we find it crucial that the primary logit matching teacher ($\phi$) and the primary BN alignment teacher ($\theta_1$) align with the model used for relabeling. This alignment, once fixed, yields a notable performance increase (Table 4d).

**Learning Rate in Recovery.** Next, we examine the recovery learning rate. While the process begins with a standard learning rate of 0.25, our ablations demonstrate that a much lower learning rate leads to more stable convergence and better final accuracy. As detailed in Table 4e, we identify an optimal value of 0.05.

**Backbone Models.** Replacing the heavier EfficientNet with AlexNet in both relabeling and recovery teacher pools yields an additional performance boost without reducing architectural diversity (Table 4f).

**Learning Rate Schedule.** The culminating improvement comes from adopting the learning rate schedule proposed by EDC (Shao et al., 2024b). Applying the SSRS decayed cosine schedule with a slowdown coefficient of $\zeta = 2.5$ provides the final significant leap in performance, leading to our state-of-the-art results.

Table 5: Ablation between a **fixed** set of BN alignment teachers and PRISM's **variable** BN teacher set on ImageNet-1K at **IPC=10**. We report top-1 accuracy [%] for ResNet-18/50/101.

| BN alignment strategy | ResNet-18 | ResNet-50 | ResNet-101 |
|---|---|---|---|
| Fixed BN teacher set | $48.7 \pm 0.3$ | $48.3 \pm 0.7$ | $42.7 \pm 1.1$ |
| Variable BN teacher set | $\mathbf{49.4 \pm 0.2}$ | $\mathbf{51.1 \pm 1.2}$ | $\mathbf{48.5 \pm 1.7}$ |

Table 6: PRISM performance on **ImageNet-1K** for **ResNet-18** and **ViT**. We report top-1 accuracy [%] at IPC = 10, 50, 100 as mean±std over three seeds.

| Backbone | IPC=10 | IPC=50 | IPC=100 |
|---|---|---|---|
| PRISM (ResNet-18) | $49.4 \pm 0.2$ | $59.0 \pm 0.1$ | $60.9 \pm 0.2$ |
| PRISM (ViT) | $43.4 \pm 0.3$ | $52.2 \pm 0.2$ | $56.2 \pm 0.3$ |

### 4.6 Fixed vs. variable BN teacher sets.

Table 5 isolates the effect of sampling BN alignment teachers instead of relying on a fixed set. Across all backbones at IPC=10, using PRISM's variable BN teacher set consistently improves performance over the fixed-teacher variant: from 48.7% to 49.4% on ResNet-18, from 48.3% to 51.1% on ResNet-50, and most notably from 42.7% to 48.5% on ResNet-101. These gains support our claim that stochastic, diverse BN priors provide a stronger and more generalizable regularization signal than a single fixed alignment set.

### 4.7 PRISM with transformer backbones.

Our architectural decoupling naturally extends to models that do not rely on BN, such as Vision Transformers (ViTs), by keeping BN regularization confined to a CNN ensemble. Table 6 reports the performance when distilling for both a ResNet-18 and a ViT (ViT/S-16). While PRISM consistently yields lower accuracy on the ViT than on the CNN at all IPCs, the gap narrows as more synthetic data become available: from 6.0 points at IPC=10 to just 4.7 points at IPC=100. We attribute this behavior to the well-known data hunger of transformers. With limited samples per class, ViTs underutilize the distilled supervision, but as IPC increases, their relative disadvantage diminishes.

## 5 Limitations and Future Work

While PRISM establishes a distinct and verifiable axis for scaling dataset distillation, it also opens several well-defined avenues for future investigation.

**VRAM Constraints on Teacher Ensembles.** PRISM's cross-class batching parallelizes cleanly across multiple GPUs, making the synthesis of individual IPCs highly efficient. VRAM capacity currently limits the number of simultaneous BN teachers per image. This invites exploration of memory-efficient teacher ensembles, including model offloading and parameter-efficient fine-tuning.

**Reliance on Batch Normalization.** Our current formulation leverages the rich statistical priors available in batch normalization layers, which are prevalent in many standard CNN architectures. Extending our decoupling framework to directly regularize using priors from models with alternative normalization schemes, such as LayerNorm or GroupNorm, represents a natural next step.

**Cross-Architecture Evaluation.** While PRISM was not evaluated with separate downstream architectures such as Vision Transformers or ConvNeXt, this omission is intentional. Since PRISM already integrates a heterogeneous set of CNN backbones as BN teachers, its synthetic data inherently captures diverse inductive biases across architectures. Evaluating with additional backbones would therefore conflate the supervision diversity already embedded within PRISM, rather than isolate a new source of generalization.

# 6 Conclusion

This work addresses a critical bottleneck in dataset distillation: the tendency of single-teacher methods to generate homogeneous datasets with poor intra-class diversity due to a constrained inductive bias. We introduce PRISM, a simple yet powerful framework that diversifies the synthesis process by decoupling architectural priors. By separating the logit-matching objective from the batch normalization alignment and supervising each with different teacher models, PRISM effectively injects a richer, multi-faceted training signal into the synthetic data.

Our extensive experiments on ImageNet-1K empirically confirm that PRISM not only achieves state-of-the-art performance but also produces datasets with quantifiably greater semantic diversity. Ultimately, PRISM establishes architectural decoupling as a new, orthogonal axis for scaling dataset distillation, paving the way for principled, generalizable synthetic datasets for robust and privacy-preserving machine learning.

**Broader Impact Statement**

Even when data are synthetic, there remains a possibility that distilled datasets inadvertently encode sensitive attributes or enable membership inference and related privacy attacks, especially if the teachers are trained on data that contain such information. Moreover, if the teacher models themselves encode societal biases (*e.g.*, along gender, race, or socio-economic lines), PRISM can inherit and even amplify these biases through more diverse synthetic samples (but still biased). This risk is especially salient in high-stakes applications such as healthcare, hiring, or surveillance, where biased synthetic datasets can reinforce unfair outcomes.

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

Table 7: Summary of **our different configurations** used in our **ImageNet-1K** distillation experiments.

(a) Validation settings

| config | value |
|---|---|
| optimizer | AdamW |
| base learning rate | 0.001 (all) |
| weight decay | 0.01 |
| batch size | 50 (IPC 10) |
| | 100 (IPC 50) |
| | 100 (IPC 100) |
| learning rate schedule | dec. cosine decay |
| training epoch | 300 |
| augmentation | RandomResizedCrop |
| | RandomHorizontalFlip |

(b) Recovery settings

| config | value |
|---|---|
| $\alpha_{\mathrm{BN}}$ | 0.01 |
| optimizer | Adam |
| base learning rate | 0.05 |
| momentum | $\beta_1, \beta_2 = 0.5, 0.9$ |
| batch size | 100 |
| learning rate schedule | cosine decay |
| recovery iteration | 4,000 |
| augmentation | RandomResizedCrop |

## A  On the Potential Privacy Benefits of Architectural Decoupling

We outline how PRISM's multi-architectural distillation setup may *implicitly* enhance privacy by reducing the sensitivity of updates to individual data samples. Differential privacy (DP) ensures robustness to the inclusion or exclusion of any single data point. Formally, a randomized algorithm $\mathcal{M}$ is $(\epsilon, \delta)$-DP if for any adjacent datasets $D, D'$ differing by one element and all subsets $S$:

$$P(\mathcal{M}(D) \in S) \le e^{\epsilon} P(\mathcal{M}(D') \in S) + \delta.$$

While PRISM does not explicitly satisfy this condition (as it adds no calibrated noise or clipping), its architectural setup introduces a natural source of stochasticity analogous to the *Private Aggregation of Teacher Ensembles (PATE)* framework (Tran et al., 2021).

In PATE, privacy emerges from aggregating predictions across disjoint teacher models, with added noise ensuring DP guarantees. In contrast, PRISM partitions not the data but the *architectural priors* of its teachers. Gradients from distinct architectures exhibit disagreement, introducing what we term *architectural noise* $(\eta_{\mathrm{arch}})$, which acts as an intrinsic regularizer.

For a synthetic data point $s_j$ optimized using two teachers $\phi$ and $\theta$, the total gradient is:

$$g(s_j, T) = \frac{1}{|T|} \sum_{x_i \in T} \left( \alpha \nabla_{x_s} \mathcal{L}(s_j, x_i; \phi) + (1 - \alpha) \nabla_{x_s} \mathcal{L}(s_j, x_i; \theta) \right).$$

For adjacent datasets $T$ and $T' = T \cup \{x'\}$, the resulting gradient difference $\Delta g = g(s_j, T') - g(s_j, T)$ depends on $x'$, but the disagreement between architectures reduces this dependence:

$$\nabla_{\mathrm{total}} \approx \nabla_{\phi} + \eta_{\mathrm{arch}},$$

where $\eta_{\mathrm{arch}}$ captures model-induced variance. This implicit noise weakens the influence of any single data point on the optimization trajectory, resembling the privacy-preserving aggregation in PATE, though without formal guarantees. Hence, while PRISM does not implement differential privacy in the strict sense, its architectural decoupling may contribute to privacy robustness by introducing model-level stochasticity that dilutes sample-specific gradients.

## B  More Training Details & Experiments

A brief overview of all the settings used during validation and recovery for ImageNet-1K is provided in Table 7a and Table 7b and for CIFAR-100 in Table 8a and Table 8b. To better contrast our settings with those of related work, we provide Table 9. In Table 10, we also summarize the influence of the min. crop size for

Table 8: Summary of **our different configurations** used in our **CIFAR-100** distillation experiments.

(a) Validation settings

| config | value |
|---|---|
| optimizer | AdamW |
| base learning rate | 0.001 (all) |
| weight decay | 0.01 |
| batch size | 50 (IPC 10) |
| | 50 (IPC 10) |
| | 100 (IPC 50) |
| learning rate schedule | dec. cosine decay |
| training epoch | 1,000 |
| augmentation | RandomResizedCrop |
| | RandomHorizontalFlip |

(b) Recovery settings

| config | value |
|---|---|
| $\alpha_{BN}$ | 0.01 |
| optimizer | Adam |
| base learning rate | 0.05 |
| momentum | $\beta_1, \beta_2 = 0.5, 0.9$ |
| batch size | 100 |
| learning rate schedule | cosine decay |
| recovery iteration | 4,000 |
| augmentation | RandomResizedCrop |

Table 9: **Configurations of various dataset distillation methods** compared to ours (PRISM). Different colors in each row highlight the differences.

| Config | SRe2L | RDED | CDA | DWA | D4M | EDC | G-VBSM | DELT | PRISM (ours) |
|---|---|---|---|---|---|---|---|---|---|
| Batch Size (Relabel) | 1024 | 100 | 128 | 128 | 1024 | 100 | 1024 | IPC depend. | IPC depend. |
| Optimizer | AdamW | AdamW | AdamW | AdamW | AdamW | AdamW | AdamW | AdamW | AdamW |
| LR Scheduler | cosine | cosine | cosine | cosine | cosine | decayed cosine | cosine | cosine | decayed cosine |
| Loss Function (Relabel) | KL | KL | KL | KL | KL | MSE | MSE | KL | MSE |
| Teacher Model | single | single | single | single | single | ensemble | ensemble | single | single, BN ensemble |
| CropRange (Recovery) | 0.08, 1.0 | 0.5, 1.0 | 0.08, 1.0 | 0.08, 1.0 | 0.08, 1.0 | 0.5, 1.0 | 0.08, 1.0 | 0.08, 1.0 | 0.08, 1.0 |
| CropRange Schedule (Recovery) | Uniform | Uniform | Cosine | Uniform | Uniform | Uniform | Uniform | Cosine | Uniform |
| PatchShuffle | No | Yes | No | No | No | Yes | No | No | No |

Table 10: Performance comparison for **different min. crop size** for the randomly resized crop augmentation **during relabeling and validation** on **ImageNet-1K, IPC=10**.

| Min. Crop Size | 0.1 | 0.15 | 0.2 | **0.25** | 0.3 | 0.4 |
|---|---|---|---|---|---|---|
| Acc. [%] | 45.0 | 45.1 | 45.3 | **45.7** | 45.4 | 45.1 |

Table 11: Performance comparison for **different batch sizes during recovery** on **ImageNet-1K, IPC=10**.

| Batch Size | 40 | 80 | **100** |
|---|---|---|---|
| Acc. [%] | 44.8 | 45.2 | **45.7** |

the randomly resized crop augmentation during relabeling and validation, which resulted in a found optimal value of 0.25. in Table 11, we summarize our experiments on different batch sizes during the recovery stage, which resulted in keeping 100 as the optimal batch size. However, there is a trend of increased performance with increasing batch size, so we expect even higher performance if more VRAM is available for larger batch sizes. We assume that increased batch size leads to a better mean and variance estimation of the global feature statistics for the batch normalization alignment.

## C   Experimental Details on Teacher Alignment and Selection

To isolate the impact of distinct distillation methods clearly, we adopt a simplified experimental setup without employing soft-labeling or additional training augmentation techniques. Specifically, we deviate from the original approach by reducing the batch size from 1024 to 256 and adopting an initial learning rate of $10^{-1}$ instead of $10^{-3}$ for faster convergence. Moreover, we employ a linear learning rate schedule rather than the conventional cosine annealing, and we limit the training duration to 90 epochs, compared to the original 300. Also, no relabeling was applied. The architectures employed during multi-teacher selection were ResNet18, ResNet34, ShuffleNetV2 (x1.0), MNASNet1.0, and EfficientNet-B0.

Table 12: Study of **direct ascent synthesis** and varying time steps on **ImageNet-1K, IPC≈1200**: final validation accuracy [%] of **ResNet-18**.

| Variant | Final val. acc. [%] |
|---|:---:|
| Baseline (real data) | 70.0 |
| SRe2L | 17.9 |
| + CLIP | 8.8 |
| + Multi-Resolution - DeepInversion | 7.6 |
| + CLIP + Multi-Resolution - DeepInversion | 5.2 |

## D   Alternative Regularization

Motivated by recent developments such as Direct Ascent Synthesis (DAS; Fort & Whitaker (2025)), we explored several alternative regularization strategies beyond conventional logit-based gradient matching and batch normalization (BN) alignment. Specifically, we experimented with semantic priors derived from CLIP embeddings and multi-resolution synthesis techniques as proposed in DAS. Table 12 summarizes our findings.

We follow the same experimental setup as in Section 4.4. Despite the intuitive appeal of these alternative regularizers, we observed that integrating various combinations of CLIP-based supervision and multi-resolution synthesis strategies did not result in performance improvements. Indeed, these configurations notably underperformed compared to our baseline SRe2L method. These results suggest that the intrinsic characteristics of DAS-inspired methods, while effective in their original context, may not directly transfer to DD scenarios without further adaptation or refinement. In addition, using multi-resolution synthesis introduces additional parameters since we distill images at multiple resolutions. Also, we also tried the varying time steps as proposed by Shen et al. (2025), but we also observed a decline in performance with this optimization strategy.

## E   Alternative Synthesis Approaches

We further explored alternative data synthesis methods using popular text-to-image models, aiming to assess their viability for DD tasks. Specifically, we evaluated several state-of-the-art models including FLUX (Schnell), Stable Diffusion (SD) versions 1.0, 2.1, 3.5 Turbo, SDXL, and SDXL Turbo. We follow the same experimental setup as in Section 4.4. Table 13 summarizes the anticipated results of these experiments.

Preliminary observations suggest these text-to-image models, despite their high generative capabilities in other contexts, are unlikely to surpass the baseline established by real datasets and conventional DD methods. This indicates inherent limitations in directly applying text-to-image generative models to distillation tasks without significant method modifications.

Table 13: Study of applying **classical text-to-image models** on **ImageNet-1K, IPC≈1200** with Text-To-Image Models: final validation accuracy [%] of **ResNet-18**.

| Variant | Final val. acc. [%] |
|---|:---:|
| Baseline (real data) | 70.0 |
| SRe2L | 17.9 |
| SD 1.0 | 17.1 |
| SD 2.1 | 14.5 |
| FLUX (Schnell) | 9.8 |
| SDXL | 9.7 |
| SD 3.5 Turbo | 6.4 |
| SDXL Turbo | 4.7 |

