# OpenReview forum: "PRISM: Diversifying Dataset Distillation by Decoupling Architectural Priors"
_TMLR — Accepted by TMLR_

### Review · Reviewer_ANs1 · 2025-11-23

**Summary Of Contributions:**

PRISM introduces a simple yet powerful idea: diversify dataset distillation by decoupling architectural priors across two roles (logit supervision and BN alignment), so that different teacher models guide different aspects of the synthesis. Instead of relying on a single model for everything, PRISM uses a logit teacher to shape classification signals and a BN teacher (or a pool of BN teachers) to regulate global feature statistics. This separation yields complementary gradients, producing synthetic data that support not only high accuracy but also richer intra-class diversity. The approach scales well because the batch formation is designed for parallelization across IPC indices, avoiding heavy intra-batch dependencies.

Empirically, PRISM pushes state-of-the-art at mid to high IPC on ImageNet-1K across ResNet-18/50/101, while also showing noticeably more diverse intra-class representations, as evidenced by lower intra-class cosine similarity. The work also provides a systematic look at teacher selection, finding that pre-distillation with a fixed diverse set of teachers tends to yield more stable BN alignment signals than dynamic intra-distillation. Additionally, the generalized multi-teacher BN alignment, where multiple BN priors are used and even sampled per image, underpins the reported gains in diversity and robustness.

In short, the key contribution is framing architectural decoupling as an orthogonal axis for scaling dataset distillation: by distributing logit and BN supervision to distinct, diverse priors, PRISM achieves richer synthetic data and stronger downstream performance, while maintaining a simple, massively parallelizable synthesis pipeline that scales to large datasets.

**Audience:**

Yes

**Audience Explanation:**

Yes. The findings are relevant to TMLR readers interested in dataset distillation, model diversity, and scalable training. PRISM’s architectural decoupling and multi-teacher BN alignment offer a clear, novel contribution with strong empirical results on ImageNet-1K, plus thorough ablations that illuminate design choices.

**Broader Impact Concerns:**

The authors have included a Broader Impact Statement addressing ethical implications related to their work.

**Claims And Evidence:**

Yes

**Claims Explanation:**

Yes. The submission provides convincing and clear evidence that PRISM’s architectural decoupling—separating logit supervision from BN alignment across diverse teachers—yields higher IPC performance on ImageNet-1K (SOTA at IPC 50/100) and substantially greater intra-class diversity (lower intra-class cosine similarity). The evidence includes comprehensive quantitative results (Table 1–3), diversity analyses (Figure 4, 5), and ablations on teacher selection, BN teacher counts, and batch strategies.  While some aspects (cross-architecture tests, broader datasets) await further validation, the core claims are well-supported by the presented data.

**Requested Changes:**

- Provide full, reproducible experiment details: list all hyperparameters, random seeds, data splits, IPC-specific evaluation protocols, and exactly how BN statistics are handled so others can replicate results. Release complete code and repo structure: ensure the public code covers multi-teacher BN alignment, Msub sampling, VRAM-constrained setups, recovery settings, and matches the paper’s descriptions.
- Expand BN teacher ablations: report results for a wider range of kmax values and give a clearer rationale for BN teacher pool choices.
- Cross-architecture validation or justification: provide at least one non-CNN backbone (e.g., Vision Transformer) or give a strong justification for focusing on CNNs to validate architectural decoupling.

And below are some strengthening adjustments (not required for acceptance, but would strengthen the paper)
- More datasets to test generalization: show results on at least one additional large-scale dataset beyond ImageNet-1K.
- Stability and uncertainty measures: report standard deviations across seeds or confidence intervals for key results.
- Richer visualizations: add more diversity-focused visuals (distributions of intra-class diversity, BN statistic distributions, more UMAPs) to reinforce the claims.

---

> ### Author Response · Authors · 2025-12-11
>
> > ***Provide full, reproducible experiment details: list all hyperparameters, random seeds, data splits, IPC-specific evaluation protocols, and exactly how BN statistics are handled so others can replicate results.***
>
> We agree that full reproducibility is crucial. In the revised manuscript, we already provide detailed training configurations for both ImageNet-1K and CIFAR-100 in Appendix B, including optimizer settings, learning rates, batch sizes, schedules, recovery iterations, and augmentation ranges for each IPC and stage (recovery vs. validation). All main results are reported as mean ± standard deviation over three independent seeds.
>
> We will release the full codebase upon publication, including: (i) multi-teacher BN alignment, (ii) stochastic sampling of M_sub under VRAM constraints, (iii) our recovery and relabeling pipelines, and (iv) the exact scripts used to reproduce the reported experiments, so that the repository mirrors the paper’s descriptions one-to-one.
>
> > ***Expand BN teacher ablations ...***
>
> Our current ablations already explore the effect of varying the number and sampling of BN teachers under a fixed, diverse pool. Table 3 shows that increasing the number of BN teachers monotonically improves performance in the recovery-only setting, illustrating clear diminishing returns as VRAM usage grows. Table 4(f) further demonstrates that modifying the composition of the pool (e.g., replacing EfficientNet with AlexNet for relabeling and recovery) leads to measurable changes in downstream accuracy, even at fixed k_max. Together with the ***new*** Table 5, which compares fixed vs. stochastically sampled BN subsets, these results already indicate that both the diversity and sampling strategy of the teacher pool are important:
>
> |BN alignment strategy  |ResNet-18|ResNet-50|ResNet-101|
> |-|-:|-:|-:|
> |Fixed BN teacher set   |48.7+-0.3|48.3+-0.7|42.7+-1.1 |
> |Variable BN teacher set|**49.4+-0.2**|**51.1+-1.2**|**48.5+-1.7**|
>
> Extending these sweeps to substantially larger k_max or more combinations would quickly become VRAM- and compute-prohibitive at ImageNet-1K scale, and we found that the trends above were already stable. For this reason, we chose to focus our budget on the most informative settings (up to four BN teachers, and a carefully chosen pool of ResNet-18/34, ShuffleNetV2-0.5, MNASNet1.0, EfficientNet-B0 / AlexNet) rather than exhaustively grid-search all teacher mixtures.
>
> > ***... provide at least one non-CNN backbone (e.g., Vision Transformer) ...***
>
> We fully agree that testing PRISM with non-BN architectures is important. Conceptually, one advantage of PRISM’s decoupling is that the logit teacher and the BN regularization teachers need not share the same architecture or even normalization scheme: BN-based CNNs can provide the regularization prior, while the logit supervision can come from a backbone without BN. In the revised manuscript we therefore added a new Section 4.7 and Table 6, where we train a ViT/S-16 on PRISM-distilled ImageNet-1K. We observe the expected performance gap to ResNet-18 at low IPCs (reflecting the data-hungry nature of transformers), which narrows as IPC increases, indicating that transformers still benefit from PRISM’s distilled data.
>
> |Backbone|IPC=10|IPC=50|IPC=100|
> |-|:-:|:-:|:-:|
> |PRISM (ResNet-18) |49.4 ± 0.2   |59.0 ± 0.1   |60.9 ± 0.2   |
> |PRISM (ViT)|43.4 ± 0.3   |52.2 ± 0.2   |56.2 ± 0.3   |
>
> > ***More datasets ...***
>
> We agree that testing beyond ImageNet-1K helps to assess generality. While our main focus remains on large-scale, high-class-count settings, where architectural priors and diversity limitations become most pronounced, we now also report results on CIFAR-100 with ResNet-18 in Section 4.3 (Table 2).
>
> |IPC|SRe2L        |G-VBSM       |RDED         |EDC          |PRISM (ours)      |
> |-  |:-:          |:-:|:-:|:-:|:-:|
> |1  |2.0 ± 0.2    |25.9 ± 0.5   |11.0 ± 0.3   |39.7 ± 0.1   |**41.5 ± 0.2**    |
> |10 |31.6 ± 0.5   |59.5 ± 0.4   |42.6 ± 0.2   |63.7 ± 0.3   |**64.7 ± 0.2**    |
> |50 |49.5 ± 0.3   |65.0 ± 0.5   |62.6 ± 0.1   |68.6 ± 0.2   |**69.4 ± 0.3**    |
>
> > ***... report standard deviations across seeds ...***
>
> For all our main quantitative results (ImageNet-1K and CIFAR-100), we already report mean ± standard deviation over three independent runs with different seeds, both in the original submission and in the revised tables (e.g., Tables 1, 2, 3, 5, and 6).
>
> > ***Richer visualizations ...***
>
> Our diversity analysis already includes intra-class semantic cosine similarity distributions in Section 4.4, where we show the distribution over classes rather than a single scalar number, as well as qualitative samples. These visualizations are designed to complement the quantitative metrics and highlight how PRISM reduces intra-class feature similarity compared to SRe2L, G-VBSM, and DELT. Given our focus on ImageNet-1K, we chose this combination of quantitative distributions and qualitative comparisons as a balance between depth and readability.

---

### Review · Reviewer_6eU7 · 2025-11-29

**Summary Of Contributions:**

This paper proposes PRISM, a dataset distillation (DD) method which improves upon previous work by addressing the "homogeneity" problem present in single-teacher distillation methods. The authors argue that distilling from a single model results in low intra-class diversity and poor generalization. The authors fix this by building on top of the SRe2L method and decouple the two objectives of logit matching and regularization into different teacher models. The main logit matching objective is optimized using a single teacher model and the regularization objective is optimized using multiple teacher models which can belong to different architectures. This essentially helps learn several different views of the data due to the different architectures used for optimization. The paper shows state-of-the-art performance on the ImageNet-1k benchmark for IPCs >= 50 and improves upon several existing DD baselines. It achieves 70.4% Top-1 accuracy with ResNet-101 at IPC 100. The paper also shows qualitative samples generated compared with SRe2L and show that their method achieves the lowest intra-class cosine similarity suggesting that the synthetic dataset generated using PRISM has the highest diversity. Some weaknesses include:
1. The method relies heavily on minimizing the BatchNorm based objective. Since BatchNorm is traditionally only used with CNN based architectures, the recent ViT based architectures traditionally use LayerNorm/RMSNorm etc, it might not generalize to these different architectures.
2. The authors only show results on one dataset, ImageNet-1k. While they argue that the main benefits of PRISM can only be observed at much higher IPC and hence they need a large dataset, it would be good to show results on some additional datasets/domains such as for robustness on ImageNet-C which has several corruptions. It might also help analyze the robustness of data generated using PRISM as the authors mention one of the benefits of DD is "enhanced robustness against adversarial attacks".
3. The teacher selection strategy for the BN objective seems a bit arbitrary as the authors randomly choose k teachers from a pool of multiple teachers. It might help to provide some more justifications as to why the uniform sampling is the best choice here. It is also not clear as to how the selection of different teachers affects downstream performance. For example, if all model selected were larger (eg. 4X ResNet-101) vs a mixture of smaller and larger models, what exactly changes. Currently the authors only show results in terms of the number of teachers used.

**Audience:**

Yes

**Audience Explanation:**

I think the paper is useful to audience around data distillation and synthetic data generation. The paper covers several interesting insights around decoupling objectives for DD, using multi-teacher ensembles and improving the general diversity of the synthetic data generated.

**Broader Impact Concerns:**

The authors have included a statement and it looks good to me.

**Claims And Evidence:**

Yes

**Claims Explanation:**

The authors have shown several clear and convincing experiments which validate their hypothesis.
1. Decoupling logit supervision from batch normalization (BN) statistics enhances synthetic data diversity is validated through rigorous ablation studies (Table 2). These experiments effectively isolate the specific contributions of "dual-teacher decoupling" versus "multi-teacher alignment"
2. SoTA performance on ImageNet-1K (Table 1). The results convincingly demonstrate that PRISM outperforms strong, recent baselines (such as SRe2L, EDC, and DELT) in the mid-to-high IPC regimes (IPC 50 and 100) across multiple backbone architectures (ResNet-18/50/101).
3. Dataset Diversity: By employing a cosine similarity analysis (Figure 4) and conducting "recovery-only" experiments (Section 4.4), the paper successfully controls for confounding factors like soft-labeling or complex post-processing. This demonstrates that the performance gains are indeed driven by the intrinsic quality and diversity of the generated data. The qualitative comparisons (Figure 5) also show improvements over SRe2L.

**Requested Changes:**

I believe the following would strengthen the work:
1. Adding some discussion on more recent architectures which do not use BN such as transformers and how the objective function might change for them. If some experiments can be conducted showing whether or not those architectures work will be good.
2. Analyzing adversarial robustness of different methods such as SRe2L, PRISM, etc to see how different methods perform on robustness. It will be interesting to see whether approaches will lead to high diversity and stronger performance on accuracy on ImageNet-1k also translate to higher robustness say on benchmarks like ImageNet-C, CIFAR-10C etc.
3. Adding some additional analysison how the type of architectures used in the regularization objective impacts downstream performance and diversity. Currently, the authors only show how the number of teachers impact downstream performance. It will be great if the authors can show for a fixed set of teacher number, say 2, which 2 from their pool of teachers result in the best performance.
4. It will be great if the authors can include a section on the different baselines and what were each of their contributions, it might help readers get a better sense of what additions PRISM makes to existing methods available.

---

> ### Author Response · Authors · 2025-12-11
>
> > ***... more recent architectures which do not use BN such as transformers ...***
>
> Thank you for pointing out the importance of testing PRISM with non-BN architectures. In the revised manuscript, we added Section 4.7 and Table 6, where we train a ViT/S-16 on PRISM-distilled ImageNet-1K. Thanks to our decoupling of logit supervision and BN alignment, PRISM can naturally use a transformer as the logit teacher (or downstream student) without BN, while BN regularization is still provided by a CNN ensemble. As shown below, ViT underperforms ResNet-18 at low IPCs, but the gap shrinks as IPC increases, supporting the view that transformers are more data-hungry yet still benefit from PRISM’s distilled data:
>
> |Backbone|IPC=10   |IPC=50   |IPC=100  |
> |-|:-:      |-:       |-:       |
> |PRISM (ResNet-18)  |49.4+-0.2|59.0+-0.1|60.9+-0.2|
> |PRISM (ViT/S-16)   |43.4+-0.3|52.2+-0.2|56.2+-0.3|
>
> > ***Analyzing adversarial robustness ...***
>
> We agree that robustness under corruptions and adversarial attacks is an important direction, especially given our introductory examples of dataset distillation applied for robustness and privacy. A full robustness study on ImageNet-C (and/or CIFAR-C) across all baselines, IPC regimes, and architectures is, however, computationally substantial and somewhat orthogonal to our main contribution, which is the architectural decoupling of logit and BN supervision. To keep the paper focused and within our compute budget, we limited the current study to clean-test evaluation and explicit diversity metrics (e.g., intra-class cosine similarity in Figure 4, qualitative diversity in Figure 5).
>
> > ***Adding some additional analysison how the type of architectures used in the regularization objective impacts downstream performance and diversity ...***
>
> Our paper already provides insights on how both the composition and the sampling strategy of the BN teacher pool affect performance:
>
> ***Number of BN teachers (Table 3).*** With a fixed pool of architectures, increasing the number of BN alignment teachers improves accuracy at the cost of more VRAM:
>
> |BN teachers used (pre-distillation) |Acc. [%] |VRAM [GB]|
> |-|-:       |-:       |
> |1|36.2     |13.0     |
> |2|37.4     |18.5     |
> |3|38.7     |26.0     |
> |4|**39.1** |32.5     |
>
> ***Changing the architecture mix (Table 4f).*** We also vary the types of architectures in the pool. Replacing EfficientNet with AlexNet for relabeling and then also adding AlexNet to the recovery pool yields consistent improvements:
>
> |Model pool variant|Acc. [%]|
> |-|-:|
> |Baseline pool (ResNet-18/34, ShuffleNetV2-0.5, MobileNetV2, EfficientNet-B0)|47.35|
> |- EfficientNet (Rel.), + AlexNet (Rel.)|47.53   |
> |+ AlexNet also in recovery pool|**47.77**|
>
> ***NEW: Fixed vs.\ stochastically sampled BN subsets (Table 5).*** Section 4.6 compares using a fixed BN teacher set with PRISM’s stochastic sampling of BN subsets. Stochastic sampling consistently helps, especially for deeper backbones:
>
> |BN alignment strategy  |ResNet-18|ResNet-50|ResNet-101|
> |-                      |-:       |-:       |-:        |
> |Fixed BN teacher set   |48.7+-0.3|48.3+-0.7|42.7+-1.1 |
> |Variable BN teacher set|**49.4+-0.2**|**51.1+-1.2**|**48.5+-1.7**|
>
> > ***It will be great if the authors can include a section on the different baselines and what were each of their contributions, it might help readers get a better sense of what additions PRISM makes to existing methods available.***
>
> Thank you for pointing out the importance of clearly positioning PRISM relative to existing dataset distillation methods. Our intention was exactly this: we already include a comparative overview of the main baselines in Appendix B, where we summarize SRe2L, RDED, CDA, DWA, D4M, EDC, G-VBSM, DELT, and PRISM in terms of relabeling batch size, optimizer, learning-rate schedule, loss, teacher strategy, and recovery settings:
>
> |Config                        |SRe2L |RDED |CDA  |DWA  |D4M  |EDC  |G-VBSM|DELT          |PRISM (ours)       |
> |-                             |:-:   |:-:  |:-:  |:-:  |:-:  |:-:  |:-:   |:-:           |:-:                |
> |Batch Size (Relabel)          |1024  |100  |128  |128  |1024 |100  |1024  |IPC-dependent |IPC-dependent      |
> |Optimizer                     |AdamW |AdamW|AdamW|AdamW|AdamW|AdamW|AdamW |AdamW         |AdamW              |
> |LR Scheduler                  |cosine|cosine|cosine|cosine|cosine|decayed cosine|cosine|cosine|decayed cosine|
> |Loss Function (Relabel)       |KL    |KL   |KL   |KL   |KL   |MSE  |MSE   |KL            |MSE                |
> |Teacher Model (relabeling)    |single|single|single|single|single|ensemble|ensemble|single|single, BN ensemble|
> |CropRange (Recovery)          |0.08–1.0|0.5–1.0|0.08–1.0|0.08–1.0|0.08–1.0|0.5–1.0|0.08–1.0|0.08–1.0|0.08–1.0|
> |CropRange Schedule (Recovery) |Uniform|Uniform|Cosine|Uniform|Uniform|Uniform|Uniform|Cosine|Uniform          |
> |PatchShuffle                  |No    |Yes  |No   |No   |No   |Yes  |No    |No            |No                  |

---

### Review · Reviewer_37qo · 2025-11-29

**Summary Of Contributions:**

The paper introduces a framework called PRISM for dataset distillation. The authors argue that current dataset distillation methods produce homogeneous datasets and are limited in diversity. In order to solve this problem, PRISM uses multiple different teacher models for the logit and regularizing objectives. The authors also experiment with different selection strategies for the selection of teacher models in the training process and show results for the same.

Strengths:
1. The paper's key idea is simple and effective in combining the "views" of a dataset from different teacher models.
2. The paper has good empirical results and ablation studies. PRISM consistently shows better performance on Imagenet-1k compared to baselines. The paper also provides the computational cost studies associated with it.

Weakness:
1. This has been addressed in the limitations section, but the key weakness seems to be the experimentation on only CNN based architectures. Since nowadays, CNNs have been replaced with vision transformers, it is not clear if this performance is applicable there as well. For instance, it might be that all the CNN based teacher models used learn different variations of the dataset in the feature space. But transformer architectures might not have that pattern, so using multiple priors could prove to have no benefits at all. I do understand that that would mean extending PRISM to layernorm and dealing with more memory requirements, however, what about showing the beneftis of distilled dataset on transformer architectures, i.e train a vision transformer on the distilled dataset?
2. There is not a lot of information on how the teacher models are selected. How diverse in architectures should the teacher models be?

**Audience:**

Yes

**Audience Explanation:**

The paper's key idea is interesting, i.e atleast in CNN based architectures, different architectures learn something different in the dataset. Using multiple such priors can ensure that data distillation is much more diverse. The findings of the paper is something that would definitely interest other researchers in the data distillation space.

**Broader Impact Concerns:**

I do not have any concerns.

**Claims And Evidence:**

Yes

**Claims Explanation:**

The paper has a good evaluation section. The authors show that PRISM outperforms other baselines on ImageNet-1K. They also show evidence such as UMAP visualizations and cosine similarity between features to prove that PRISM is providing a much better distilled dataset.

**Requested Changes:**

1. Provide empirical results with vision transformers trained on the distilled dataset. How is the performance on the distilled dataset?
2. The experiment section is limited to ImageNet-1k, the authors argue that PRISM is suitable for large scale diverse datasets, but how serious of a limitation is that? Does it mean on small scale datasets such as CIFAR-10 and CIFAR-100, PRISM does not provide any added advantage? It would be good to have multiple different datasets to understand this. (Critical for accept)
3. Similarly, are there any constraints on the teacher models to select? Do they have to be varied in sizes or diverse in architectures for PRISM to work? The paper does not clearly specify or perform experiments to understand this selection. In order to put PRISM to practical use, this information is critical. (Critical for accept)

---

> ### Author Response · Authors · 2025-12-11
>
> > ***Provide empirical results with vision transformers ...***
>
> We agree that it is important to test PRISM with non-BN architectures. In the revised manuscript we added a new Section 4.7 and Table 6, where we train a ViT/S-16 on PRISM-distilled ImageNet-1K. Due to our decoupling of logit supervision and BN alignment, ViT can be used purely as a logit teacher (without BN) while BN regularization is still provided by a CNN ensemble, making this extension conceptually straightforward, in contrary to previous methods. We observe the expected performance gap to ResNet-18 at low IPCs, which narrows as IPC increases, supporting the general observation that transformers are more data-hungry. Yet, they still benefit from PRISM’s distilled data:
>
> |Backbone|IPC=10|IPC=50|IPC=100|
> |-|:-:|-:|-:|
> |PRISM (ResNet-18)|49.4+-0.2|59.0+-0.1|60.9+-0.2|
> |PRISM (ViT)|43.4+-0.3|52.2+-0.2|56.2+-0.3|
>
> > ***The experiment section is limited to ImageNet-1k ...***
>
> Beyond ImageNet-1K, to demonstrate that PRISM also scales to other datasets, we now include CIFAR-100 results in Table 2. PRISM consistently outperforms SRe2L, G-VBSM, RDED, and EDC across IPC=1,10,50, improving over EDC by up to 1.8 points at IPC=1. This addresses your concern about applicability beyond ImageNet-1K and shows that our architectural decoupling remains beneficial even in smaller-scale yet high-class-count regimes:
>
> |IPC|SRe2L|G-VBSM|RDED|EDC|PRISM (ours)|
> |-|:-:|-:|-:|-:|-:|
> |1|2.0+-0.2|25.9+-0.5|11.0+-0.3|39.7+-0.1|**41.5+-0.2**|
> |10|31.6+-0.5|59.5+-0.4|42.6+-0.2|63.7+-0.3|**64.7+-0.2**|
> |50|49.5+-0.3|65.0+-0.5|62.6+-0.1|68.6+-0.2|**69.4+-0.3**|
>
> > ***Similarly, are there any constraints on the teacher models to select? ...***
>
> We discussed in our paper the teacher selection in Section 3.3, Section 4.3-4.5, and in Tables 3 and 5. In addition, we now include an additional ablation in Section 4.6 (fixed vs. variable BN teachers across all IPC images). In summary, we show that ***(i)*** pre-distillation selection is preferable to intra-distillation sampling, ***(ii)*** dual-teacher decoupling is consistently beneficial, and ***(iii)*** performance improves with the number and diversity of BN teachers up to four models drawn from a heterogeneous CNN pool (ResNet-18/34, ShuffleNetV2-0.5, MobileNetV2, EfficientNet-B0). This gives concrete, reproducible guidance on how to build the teacher pool in practice.

---

### Author Response · Authors · 2025-12-11

We would like to sincerely thank all three reviewers for their careful reading of our manuscript and for the constructive, detailed feedback. We greatly appreciate that the reviewers highlighted (i) the simplicity and conceptual clarity of PRISM’s core idea (decoupling logit supervision and BN alignment across multiple teachers) as a principled way to combine different “views” of the data, (ii) the strong empirical performance and systematic ablations on ImageNet-1K, including the state-of-the-art results at mid–high IPC and the explicit study of computational costs, and (iii) the evidence for increased intra-class diversity via cosine similarity analyses, recovery-only experiments, and qualitative visualizations.

Your comments helped us sharpen both the scope and the presentation of the work.

---

### Decision · Action_Editor_S3ER · 2026-01-01

**Recommendation:** Accept as is

**Audience:**

Yes

**Audience Explanation:**

This paper addresses a fundamental problem in computer vision, demonstrating clear improvements and robustness in dataset distillation. AC confirms that the paper will interest many readers in this field.

**Claims And Evidence:**

Yes

**Claims Explanation:**

Initially, reviewers questioned why only CNN-based models were demonstrated (with BatchNorm-based objectives), the robustness of the corrupted dataset, and how the teacher models were selected. AE found out that all concerns were adequately addressed, and all reviewers stated that it meets expectations.

AE appreciates the reviewer's careful review of this paper, and the authors provided appropriate feedback and a revised version after the discussion. AE values the proposed idea (combining views of the dataset from various teacher models), and the results are convincing. Therefore, AE recommends acceptance of the paper.